# Solar UV Irradiance in a Changing Climate: Trends in Europe and the Significance of Spectral Monitoring in Italy

**Ilias Fountoulakis** [1,*], **Henri Diémoz** [1,2], **Anna-Maria Siani** [3], **Gudrun Laschewski** [4], **Gianluca Filippa** [1], **Antti Arola** [5], **Alkiviadis F. Bais** [6], **Hugo De Backer** [7], **Kaisa Lakkala** [5,8], **Ann R. Webb** [9], **Veerle De Bock** [7], **Tomi Karppinen** [8], **Katerina Garane** [6], **John Kapsomenakis** [10], **Maria-Elissavet Koukouli** [6] and **Christos S. Zerefos** [10,11,12]

[1]  Aosta Valley Regional Environmental Protection Agency (ARPA), 11020 Saint-Christophe, Italy; h.diemoz@arpa.vda.it (H.D.); g.filippa@arpa.vda.it (G.F.)
[2]  Institute of Atmospheric Sciences and Climate, National Research Council, 00185 Rome, Italy
[3]  Physics Sapienza Università di Roma, 00185 Rome, Italy; annamaria.siani@uniroma1.it
[4]  Deutscher Wetterdienst, Research Centre Human Biometeorology, Stefan-Meier-Str. 4, 79104 Freiburg, Germany; gudrun.laschewski@dwd.de
[5]  Finnish Meteorological Institute, Climate Research Programme, 70211 Kuopio, Finland; antti.arola@fmi.fi (A.A.); kaisa.lakkala@fmi.fi (K.L.)
[6]  Laboratory of Atmospheric Physics, Aristotle University of Thessaloniki, 54124 Thessaloniki, Greece; abais@auth.gr (A.F.B.); agarane@auth.gr (K.G.); mariliza@auth.gr (M.-E.K.)
[7]  Royal Meteorological Institute of Belgium, Ringlaan 3, 1180 Brussels, Belgium; Hugo.DeBacker@meteo.be (H.D.B.); Veerle.DeBock@meteo.be (V.D.B.)
[8]  Finnish Meteorological Institute, Space and Earth Observation Centre, 99600 Sodankylä, Finland; tomi.karppinen@fmi.fi
[9]  Centre for Atmospheric Sciences, Department of Earth and Environmental Sciences, University of Manchester, Manchester M13 9PL, UK; ann.webb@manchester.ac.uk
[10]  Research Centre for Atmospheric Physics and Climatology, Academy of Athens, 10680 Athens, Greece; johnkaps@geol.uoa.gr (J.K.); zerefos@geol.uoa.gr (C.S.Z.)
[11]  Navarino Environmental Observatory (NEO), Costa Navarino, 24001 Messinia, Greece
[12]  Center for Environmental Effects on Health, Biomedical Research Foundation of the Academy of Athens, 11527 Athens, Greece
[*]  Correspondence: i.fountoulakis@arpa.vda.it

**Abstract:** Review of the existing bibliography shows that the direction and magnitude of the long-term trends of UV irradiance, and their main drivers, vary significantly throughout Europe. Analysis of total ozone and spectral UV data recorded at four European stations during 1996–2017 reveals that long-term changes in UV are mainly driven by changes in aerosols, cloudiness, and surface albedo, while changes in total ozone play a less significant role. The variability of UV irradiance is large throughout Italy due to the complex topography and large latitudinal extension of the country. Analysis of the spectral UV records of the urban site of Rome, and the alpine site of Aosta reveals that differences between the two sites follow the annual cycle of the differences in cloudiness and surface albedo. Comparisons between the noon UV index measured at the ground at the same stations and the corresponding estimates from the Deutscher Wetterdienst (DWD) forecast model and the ozone monitoring instrument (OMI)/Aura observations reveal differences of up to 6 units between individual measurements, which are likely due to the different spatial resolution of the different datasets, and average differences of 0.5–1 unit, possibly related to the use of climatological surface albedo and aerosol optical properties in the retrieval algorithms.

**Keywords:** solar UV radiation; Italy; Europe; ozone; aerosols; clouds; OMI; UV forecast

## 1. Introduction

The ultraviolet (UV) band of the electromagnetic spectrum extends from 100 to 400 nm and is divided into three sub-regions: the UV-C (100–280 nm), the UV-B (280–315 nm), and the UV-A (315–400 nm). Only a small fraction (9.3%) of the electromagnetic radiation emitted by the Sun is in the UV spectral region [1] as most of it is attenuated by the Earth's atmosphere as it propagates toward the surface.

Despite the small amount of UV radiation that finally reaches the Earth's surface, its biological significance is exceptional [2]. UV radiation triggers and/or drives photochemical and photobiological processes, which are necessary for the proper functioning of ecosystems, and has strong direct or indirect effects on human health [3–21]. Living organisms have slowly adapted to the current levels of solar UV radiation through the evolution process and fast or abrupt changes impact the health and the diversity of flora and fauna [4]. Consequently, any change in the ecosystems affects human populations through their interaction with the natural environment.

UV radiation that reaches the surface of the Earth exhibits periodical changes associated with a number of different phenomena: the solar radiation that reaches the Earth's atmosphere changes by ±3% in the year as a result of periodical changes in the Earth–Sun distance while the angle between the Sun and the zenith of a particular place on Earth also changes periodically leading to corresponding changes in the radiant energy. Periodical changes in solar activity, such as the 11-year solar cycle, the 27-day apparent solar rotation, and dynamical atmospheric processes (e.g., quasi-biennial oscillation (QBO)) also induce changes in the levels of solar UV radiation that reach the Earth's surface. Living organisms have adapted to the above periodicities, which are predictable and easily modeled.

Non-periodic changes in atmospheric composition and dynamics can also affect the levels of surface UV radiation significantly. Solar UV radiation at wavelengths below 290 nm is absorbed by molecular oxygen and the ozone at the higher atmosphere and does not enter the troposphere. Atmospheric molecules scatter radiation more effectively with decreasing wavelength, thus scatter more UV than visible or infrared light. The ozone effectively absorbs UV-B radiation [22–25], allowing to only a very small fraction to reach the Earth's surface. In addition to the ozone, other gases also absorb part of the UV radiation (e.g., $SO_2$ and $NO_2$), though their impact is usually less significant than that of the ozone, either because their absorption efficiency is smaller or because they are less abundant than the ozone.

Clouds and aerosols in the troposphere also scatter (both clouds and aerosols) and absorb (aerosols) solar UV radiation. The spectral attenuation by clouds and aerosols depends on their type and characteristics [26,27]. Clouds are the most significant driver of short term variability of UV irradiance at the Earth's surface [28]. Although they usually attenuate UV radiation, under particular conditions UV irradiance can be enhanced due to the presence of clouds [29–34]. In urban environments changes in aerosols may counterbalance the effect of even extremely high or low total ozone events, and lead to erythemal irradiance above or below the climatological averages respectively [35]. Surface albedo is an additional important regulatory factor for the levels of UV irradiance, since multiple reflections between the Earth's surface and the atmospheric constituents enhance UV irradiance over highly reflective (e.g., snow- or ice-covered) terrains [36,37]. Over highly reflective terrains and under broken cloud conditions short-term enhancement can be of the order of 50% [38]. The UV radiation reaching the surface generally increases with altitude, mainly because of the decreasing atmospheric density, thus decreasing attenuation [39,40].

According to the results of many studies, long-term changes in cloudiness, ozone, surface reflectivity, and/or aerosols in the recent past have already altered the levels of UV irradiance significantly in several areas around the globe [41]. In the 1970s, increased photochemical destruction of ozone due to anthropogenic emissions of the ozone depleting substances (ODS) led to depletion

of stratospheric ozone, especially over high latitudes of both hemispheres [42–44]. Ozone depletion led to increasing UV-B radiation over the mid and high latitudes [45,46], which caused awareness in the scientific community and led to the implementation of the Montreal Protocol, which was signed in 1987. The implementation of the Montreal Protocol in 1989 was successful preventing humanity from extreme exposures to UV radiation [47–49]. Currently the first signs of ozone recovery since the mid 1990s, due to the reduced ODS emissions, have been reported over polar latitudes [50,51]. Over northern hemisphere mid-latitudes, ozone increases in the upper—but not in the lower—stratosphere have been observed in the same period [52–54]. According to Eleftheratos et al. [55] at northern high latitudes, between 55° and 70° N, the irradiances at 305 nm decreased significantly by 3.9% per decade in the period 1990–2011 mainly because of the reported ozone recovery.

Nevertheless, recent studies reporting results from mid-latitude stations of the northern hemisphere show that UV irradiance has increased in the last two decades mainly as a result of the global brightening effect, while total ozone has either remained stable or increased in the same period [56–58]. Increasing aerosols over South-East Asia have had the opposite effect resulting to decreased levels of solar UV radiation [41,59]. Changes in surface albedo and cloudiness have also affected the levels of UV radiation in the past decades, especially over sub-polar and polar regions [60–62]. For example, statistically significant negative trends of the monthly average noon erythemal and 345 nm irradiances for October, of the order of −11% to −14%, have been found for Barrow, Alaska for the period 1991–2011, which have been attributed to a statistically significant decrease in the number of days during which the land surface around the station was covered by snow. Over such high latitudes decreased ice and snow cover (i.e., decreased surface albedo) and increased cloudiness are projected to drive significant changes of UV irradiance in the future [63,64].

Due to the very complex interactions of solar UV radiation with atmospheric constituents and the features of the Earth's surface [65], modeling and predicting its changes is in many cases very uncertain [41,66,67]. Large uncertainties in the future projections (by climate models) of the main factors controlling the levels of the solar UV irradiance at the Earth's surface (especially clouds and aerosols) further increase the uncertainty in estimates of future UV changes due to the corresponding changes in climate and air quality [66]. The spectral characteristics of the interactions between UV radiation and aerosols are not yet completely understood, and are usually poorly described in radiative transfer models [41,67]. Uncertainties about the future rate of ozone recovery arise from the recent unexpected emissions of anthropogenic ODS already controlled by the Montreal protocol, as well as emissions of ODS, which have not yet been forbidden [68,69]. Climate change—induced alterations of stratospheric temperatures and circulation patterns are also expected to have a significant impact on the future levels and spatial distribution of stratospheric ozone [70–73], which in turn would affect solar UV radiation [74–79].

Continuous estimates of the levels of UV irradiance at the Earth's surface on a global scale are available for the last four decades from satellite measurements [80–83]. Although in the last years there has been significant progress in the algorithms used for the retrieval of surface UV irradiance from satellites [84,85], the retrievals are still not sufficiently accurate over mountainous sites [86,87], highly reflective terrains [88], as well as over highly polluted environments [89–91], mainly because of the use of climatological data (for e.g., surface albedo and aerosol absorption) and simplifications in the algorithms. Furthermore, satellite retrievals represent the average of a finite area covered by the satellite pixel and are not necessarily representative for each point of the pixel, especially over complex, inhomogeneous terrains [92]. Thus, under the conditions described above, comparisons between ground-based UV measurements and satellite retrievals may yield differences of 20% for clear skies and up to 50% for cloudy skies.

The accuracy of ground-based measurements is limited mainly by the characteristics of each particular instrument [93,94]. The standard uncertainty in good quality spectral UV measurements from well maintained and calibrated sensors is of the order of 5% for wavelengths above 305 nm [95,96], although it can be much lower, less than 2%, for properly designed and accurately characterized

instruments [97]. Performing continuous and high quality global spectral UV measurements is a difficult task, which demands expensive equipment and properly trained personnel. Although monitoring of the UV irradiance can be achieved using much cheaper broad-band instruments, spectral measurements have important advantages. Analyzing accurate long-term spectral UV measurements with respect to measurements of atmospheric constituents (e.g., total ozone column, aerosols, etc.) and the Earth's surface properties (e.g., surface albedo) allows the detection of trends [98,99], but also the identification of the main factors and mechanisms that drive them [26,100]. Furthermore, co-located measurements of the solar spectral UV irradiance and other atmospheric components improve our understanding regarding the complex interactions between UV radiation, atmospheric constituents, and the Earth's surface properties. In this way, analyzing spectral UV measurements contributes to the improvement of the accuracy in UV modeling and the satellite retrievals.

Spectral UV measurements can be weighted with well defined action spectra and then used for the calculation of biologically effective quantities in order to directly quantify the effect of UV radiation on biological processes [101] and human health [102,103]. A quantity that is commonly used for public information and in human health-related studies is the UV index [104]. The UV index is a metric of effectiveness of UV irradiance on causing erythema in human skin, and is calculated by dividing the erythemal irradiance (in mW/m$^2$) by 25. The erythemal irradiance is the spectral UV irradiance weighted with the erythemal action spectrum (i.e., the relative contribution of the irradiance at each wavelength to the induction of erythema in the human skin) [103,105]. The integral of erythemal irradiance over a certain time period is usually referred as the erythemal dose. Highly accurate spectral measurements can be also used for the calibration and/or the validation of broad-band and narrow-band instruments [90,97,106] used to directly measure UV index.

Due to their high cost, long-term, continuous, and accurate spectral UV measurements are available only from a limited number of stations. An investigation into the databases of the larger networks and data centers providing ground-based remote sensing measurements (the Network for the Detection of Atmospheric Composition Change (NDACC), the World Ozone and Ultraviolet Radiation Data Centre (WOUDC), and the European UV Database (EUVDB)) reveals that there are less than seventy stations around the world that have provided spectral UV measurements for relatively long periods, of a few years or more, in the last three decades. Less than half of them have provided measurements continuously for more than ten years. In Europe, measurements of the UV index and other integrated quantities are widely available, mainly from broad-band instruments [107]. There are also studies discussing the changes of the UV irradiance, even before the 1990s, based on reconstructed UV time-series from ground based or satellite retrievals of other parameters. Bilbao, et al. [108], for example, analyzed a reconstructed dataset of the erythemal irradiance for central Spain and reported a significant increase of 3.5% and 4.1% per decade for summer and autumn between 1991 and 2010. Analysis of satellite measurements also shows increasing trends of the UV-A radiation over Europe in the period 1979–2011 [109]. Systematic, continuous spectral measurements of global solar UV irradiance, which can provide more reliable information regarding the trends of UV irradiance, are available from a few European stations, in some since the early 1990s [58,110–113].

The present study provides a brief review of the results presented in recent studies reporting changes of the spectral UV irradiance and effective UV doses over Europe, with respect to changes in total ozone, surface albedo, clouds, and aerosols. The review is provided in Section 3, where long term spectral UV records from four different European stations at latitudes between 40 and 67° N are also being analyzed to provide further context. Interesting findings from the analysis of spectral UV measurements around Italy are discussed in Section 4. Thereafter, in the same section, the necessity of having continuous, accurate long-term spectral UV measurements is highlighted through the climatological analysis of the spectral datasets of Rome and Aosta, as well as comparison of the two datasets with UV estimates from the Deutscher Wetterdienst (DWD) forecast model and the ozone monitoring instrument (OMI). The methodology followed in order to analyze the data presented in Sections 3 and 4 is discussed in Section 2.

## 2. Data and Methodology

### 2.1. UV Trends in European Stations

In Section 3 of the present study, the analysis of the same dataset used in Fountoulakis et al. [56] for four different historical European stations has been updated and performed for the period 1996–2017. Ground based spectral UV measurements for the particular period have been analyzed in the context of the present study for: Uccle, Belgium (50.8° N, 4.4° E, 100 m a.s.l.), Reading, UK (51.4° N, 0.9° W, 66 m a.s.l.), Sodankylä, Finland (67.4° N, 26.6° E, 180 m a.s.l.), and Thessaloniki, Greece (40.6° N, 23.0° E, 60 m a.s.l.). The particular four stations have been chosen because they are at quite different latitudes, and provide good quality continuous spectral UV measurements at the desired wavelengths since the mid-1990s. Although measurements of the spectral UV irradiance in Reading have been performed since 1991, total ozone is available only since 2002 [60]. Thus, total ozone from the Modern-Era Retrospective analysis for Research and Applications version 2 (MERRA-2) [114] has been extracted for Reading from the Giovanni data visualization base (https://giovanni.gsfc.nasa.gov/Giovanni). Ground-based total ozone data for the remaining three stations have been downloaded from WOUDC. The spectral UV data have been either downloaded from WOUDC (for Uccle and Reading), or from the EUVDB [115] (for Sodankylä). The spectra for Thessaloniki were directly provided by the Laboratory of Atmospheric Physics of the Aristotle University of Thessaloniki, Greece (LAP-AUTH). Analytical information regarding the datasets employed and the methodology can be found in Fountoulakis et al. [56]. A brief description of the methodology is provided in the following.

The all-sky noon irradiance (±30 min around the local solar noon) at 307.5 nm and 324 nm, and the daily average total ozone was calculated for each day. The ratio between the noon irradiance at 307.5 nm and 324 nm (hereafter referred to as the 307.5/324 nm ratio) was also calculated. The long-term change of the 307.5/324 nm ratio is indicative of the relative contribution of total ozone to the changes of UV irradiance since the irradiance at 307.5 nm is strongly affected by the ozone, while the 324 nm irradiance is practically not affected by the ozone. The irradiance at 307.5 nm was chosen for the investigation of the effect of changes in total ozone, instead of the commonly used irradiance at 305 nm, because the former is less affected by measurement noise than the latter, and at the same time the effect of the ozone remains strong. Furthermore, the absorption cross section of sulfur dioxide at this particular wavelength is minimal relative to other wavelengths in the range 306–309 nm.

For each day of the year, climatological values were derived by averaging the corresponding daily values for the entire period of study. Daily anomalies (% change relative to the climatological values) were then calculated by subtracting the climatological values from the corresponding days in the entire dataset. When measurements for at least fifteen days were available for each month, the daily anomalies were averaged in order to get monthly anomalies. The periodical effects of the 11-year solar cycle and the QBO of the winds in the equatorial stratosphere have been removed using the methodology described in Zerefos et al. [57]. For that purpose, monthly means for the solar flux at 10.7 cm were downloaded from the NOAA national geophysical data centre [116], while for the QBO wind data were downloaded from the Freie Universität Berlin [117]. The monthly anomalies for ozone and irradiance were then averaged to derive annual mean anomalies. For the calculation of the annual anomalies for Sodankylä, only measurements for months April–September were taken into account to avoid high uncertainties due to the very low noon solar elevation in winter.

### 2.2. UV in Rome and Aosta

In Section 4, spectral UV measurements from the Italian stations of Rome (41.9° N, 12.5° E, 60 m a.s.l.) in central Italy, and Aosta (45.7° N, 7.4° E, 570 m a.s.l.) in the North, were analyzed and compared to each other, as well as with satellite retrievals and forecast model estimations. The UV monitoring station of Rome is located at the Solar Radiometry Observatory of Sapienza University, at a distance of about 30 km from the Tyrrhenian Sea, and is a typical urban station. The monitoring station of Aosta is located at the facilities of the Regional Environmental Protection Agency of the Aosta Valley

(ARPA Valle d'Aosta), at Aosta, Saint Christophe. The Aosta monitoring site is a semi-rural site at the North-Western Alps, inside the Aosta Valley. Tall mountains at the north and the south of the Aosta station block the horizon up to an elevation of 20° [106]. Ancillary ground based and satellite measurements of other parameters were also used and discussed further below.

UV spectra from the Brewer spectrophotometer with serial number 67 (hereafter referred to as Brewer#067), which performs measurements in the range 290–325 nm with a step of 0.5 nm and a resolution of 0.6 nm, and the Bentham DTMc300 with serial number 5541 (hereafter referred to as Bentham), which performs measurements in the range 290–400 nm with a step of 0.25 nm and a resolution of 0.5 nm were analyzed for Rome and Aosta respectively, for the period 2006–2015 (for which good quality continuous spectral measurements are available for both sites). Total ozone data from the Brewer#067 and the Brewer with serial number 66 (Brewer#066) were analyzed for Rome and Aosta respectively. UV and total ozone datasets from the two stations have all been subjected to quality control/quality assurance (QC/QA), and are traceable to international reference standards [97,106,118]. The standard uncertainty (for wavelengths above 305 nm and solar zenith angles (SZA) below 75°) of the UV spectra is estimated as 5% for Rome and 3% for Aosta [96,106]. The accuracy of the total ozone measurements from Brewer spectrophotometers is estimated to be 1% [118,119]. Spectra from the two stations have been processed using the SHICrivm algorithm [120] in order to correct them for the effect of wavelength shift, and extend the spectra up to 400 nm for Brewer #067 (in order to calculate erythemal irradiance).

The UV index has been calculated using the effective spectrum for erythema described in Webb et al. [103]. Averages of the irradiance in the range 305–310 nm (from now on referred to as 307.5 nm irradiance) and 320–325 nm (hereafter referred to as 322.5 nm irradiance) have been calculated and analyzed instead of measurements at the central wavelengths of the two bands. This way the effects of the technical characteristics of each instrument are minimized and the comparison between the irradiance at the two stations is more reliable. Increased surface albedo enhances the irradiance in the UV-B more effectively than in the UV-A region [36], while the spectral effects of clouds and aerosols vary depending on their characteristics [27,121]. However, the spectral effect of clouds, aerosols, and surface albedo is usually insignificant relative to the spectral effect of ozone, which has a much stronger effect on the 307.5 nm relative to the 322.5 nm irradiance [26,122].

Measurements performed within ±15 min from the exact local solar noon have been averaged in order to calculate the noon UV index and the noon irradiance at 307.5 and 322.5 nm. This averaging results to slightly underestimated noon irradiance, which is albeit less than 0.5% in summer and 2% in winter relative to the irradiance at the SZA of the exact local noon, as was estimated from model simulations for different atmospheric conditions. Total ozone has been averaged over the whole day. Monthly averages have been calculated for all months for which at least 15 daily measurements are available. For UV, ratios between the monthly values of the same quantities for the two sites have been also calculated. The monthly values and the ratios have been averaged over the entire period of study for each month of the year. This way, climatological values of the daily average total ozone, the noon UV index, the irradiances at 307.5 nm and 322.5 nm, and the corresponding ratios for the two sites were calculated.

In order to distinguish the effects of different geophysical parameters, the irradiances at 307.5 and 322.5 nm were also analyzed for selected SZAs. In this way the effect of different SZAs at noon over the two stations was removed and the effects of differences in cloudiness, surface albedo, and aerosol were depicted more clearly in the ratios. More specifically, daily averages of the irradiance for ±0.5° intervals corresponding to central SZAs 63–67° with a step of 1°, were calculated. Monthly mean values were again derived when at least 15 daily values were available. Then, ratios between the monthly means for the two stations, corresponding to the same SZA (each SZA from 63 to 67° with a step of 1°), were calculated. For each month, the ratios for the five different SZAs (63–67°) were averaged in order to get a single value, which was assumed to correspond to the SZA of 65°. Finally, monthly values were averaged over the 11-year period of study in order to get monthly climatological

values of the ratio. The standard deviation was calculated at the first step (when daily irradiances were averaged in order to get monthly averages) and was then transferred until the final step using classical error propagation theory.

Furthermore, radiative transfer simulations for cloudless skies were performed for the two sites, for each day of the year, for SZA = 65°. The UVSPEC radiative transfer model of the libRadtran package [123] was used for the simulations. The surface albedo was set to 0.05 (which is a typical value for snowless surfaces) and the aerosol optical depth (AOD) was set to zero. Monthly climatological values of total ozone were interpolated to each day of the year, and were then used as model inputs. A standard atmospheric profile [124] and the disort pseudospherical approximation [125] running with six streams were used. Daily values were averaged in order to get monthly means. Again ratios between the corresponding monthly means were calculated. From the modeled ratios we were able to estimate the effects of differences in the noon SZA, altitude, and the total ozone at the two stations assuming cloudless skies, zero aerosol, and snowless land surface. By comparing modeled and measured ratios at the SZA of 65° we were able to identify the overall effects of the corresponding differences in cloudiness, surface albedo, and aerosols.

Different products from the MERRA-2 [114] were extracted for Rome and Aosta from the Giovanni data visualization base (https://giovanni.gsfc.nasa.gov/Giovanni), for the time period 2006–2015 and used in order to understand the different roles of clouds and aerosols, i.e., the total cloud area fraction (TCAF) and the AOD at 550 nm. Although ground-based AOD is available for both stations [126,127], which would provide a more accurate climatology [128], the AOD from MERRA-2 was chosen since it was available for the whole period of study (2006–2015). Monthly means of AOD and TCAF were downloaded and then averaged over the period 2006–2015 in order to get the monthly climatologies. The MERRA-2 products were produced from the synergy of satellite retrievals and model simulations for a $0.5° \times 0.625°$ grid (latitude $\times$ longitude) and were extrapolated to the point of interest. Although the relationship between the TCAF and the attenuation by clouds was not necessarily linear, and the AOD at 550 nm differed from the AOD in the UV, comparison of these quantities at the two sites provides useful information regarding the seasonal patterns of the differences in the effects of clouds and aerosols.

In order to assess the effect of different surface albedo, high resolution (500 m $\times$ 500 m grid) estimates of the maximum fractional snow cover in 8-day intervals from a moderate resolution imaging spectroradiometer (MODIS) [129,130] were used. When the particular quantity was averaged over an area of 5 km around the station of Aosta, its variability is highly representative of the variability of surface albedo at the station [131]. For comparability reasons, the same averaging was performed for Rome, although the particular quantity might not be correlated with surface albedo in this case since land-surface and atmospheric conditions differed significantly between Rome and Aosta.

The differences between the magnitude and the variability of the particular parameters (TCAF, surface albedo, and AOD) at the two sites were discussed with respect to the corresponding differences in the levels of the solar UV irradiance. Since the particular satellite products are uncertain over complex terrains [132,133] (such as the Aosta Valley) and urban environments [92,134] (such as Rome), the discussion was constrained to a qualitative description of their effects, and no quantitative analysis was performed.

*2.3. Comparison of Ground-Based UV Index from Ground-Based Measurements with Forecasts and Satellite Estimates*

The all-sky UV index measured at the solar local noon in Aosta and Rome was compared with estimates from the DWD model and the OMI observations in order to estimate the magnitude of the differences and identify the main factors behind those differences. The comparison was performed for a four-year period (2012–2015).

Simulations of meteorological and radiative transfer models were merged and large-scale forecasts of the UV index were provided on a daily basis by the DWD center (https://kunden.dwd.de/uvi/index.

jsp) [135]. In particular, the outputs of the operational global icosahedral-hexagonal gridpoint model (GME) were used for forecasting until 2014. Since 2015, however, the German center has moved to the icosahedral nonhydrostatic model (ICON). The resolution is 20 km for the former and 13 km for the latter data set. As the results from the comparison of the two different models with the ground-based measurements are very similar, the daily noon UV index from the two different models was discussed as being from a single source, and was referred as the UV index from the DWD model. In the period June 2014–January 2015 malfunctions of the ozone forecasting system affected the simulations of the UV irradiance from the DWD model. Thus, data for the particular period were not included in the study. Total ozone from the DWD model for the same period was also used here.

OMI is a satellite instrument on board the NASA EOS-Aura satellite, in orbit since 2004 [136], and provides measurements of total columns of ozone and other trace gases (e.g., $SO_2$, $NO_2$, and HCHO), as well as aerosol optical properties. For the particular study, daily values of the noon UV index from the Global Level 3 OMI Surface UV Irradiance and Erythemal Dose (OMUVBd L3) product [137] were downloaded for the period 2012–2015 through the GIOVANNI interface developed by NASA (http://giovanni.gsfc.nasa.gov/giovanni/). The cloud area fraction from the same product was also downloaded. This particular product was available on a global scale with a resolution of $1° \times 1°$ ($100$ km $\times$ $100$ km). Daily averages of total ozone from the Level-3 Aura/OMI Global TOMS-Like Total Column Ozone gridded product (OMTO3e) [138], which is available on a global scale with a resolution of $0.25° \times 0.25°$, were also downloaded and compared with daily averages of total ozone measured at the two stations.

## 3. UV Trends in Europe

### 3.1. Main Findings from Recent Studies

The trends of the spectral UV irradiance since the early- or the mid-1990s, when spectral measurements are available, have been investigated in a number of recent studies. There are also studies investigating the trends of integrated UV quantities, even since the mid-1970s, based on broad-band UV measurements. In this section, the UV trends (and the factors that are mainly responsible for them) reported in the existing bibliography are discussed for European sites. A summary of the most pertinent results is provided in Table 1. For the sake of comparability, only results referring to European stations for the period between the 1990s and 2010s were included in the table. Results from studies referring to different locations and periods are also discussed in the text, even though they have not been included in Table 1. At this point it has to be clarified that the results presented in Table 1 represent the average rate of the change in UV for the full period of study at each site, and not necessarily trends. It is evident from Table 1 that the magnitude and the direction of the changes vary significantly, mainly depending on the studied quantity and the geographical location of the stations. In the following discussion, statistically significant changes refer to a significance level of 95% (*p*-value of 0.95 or larger) unless something else is specified.

**Table 1.** Long-term changes of the UV irradiance at different European stations.

| Station(s) | Latitude, Longitude, Altitude (a.s.l.) | Period | Quantity | Trend (Change % Per Decade) | Reference |
|---|---|---|---|---|---|
| Hornsund, Svalbard, Norway | 77.0° N, 15.6° E, 10 m | 1996–2016 | Daily erythemal dose | 3.5 | [139] |
| Sodankylä, Finland | 67.4° N, 26.6° E, 180 m | 1990–2014 | Irradiance at 305 nm (SZA = 64°) for April and June | −10 | [140] |
| Sodankylä, Finland | 67.4° N, 26.6° E, 180 m | 1992–2016 | Noon irradiance at 307.5 nm | −3 | [56] |
| NILU stations, Norway | Latitudes 60–79° N | 1995–2016 | Daily erythemal dose | −5 to −2 | [141] |
| Chilton, UK | 51.6° N, 1.3° W, 123 m | 1995–2015 | Daily erythemal dose | −8 * | [142] |
| Reading, UK | 51.4° N, 0.9° W, 66 m | 1992–2016 | Noon irradiance at 307.5 nm | −12 * | [56] |
| Uccle, Belgium | 50.8° N, 4.4° E, 100 m | 1991–2013 | Daily erythemal dose | +7 * | [58] |
| Uccle, Belgium | 50.8° N, 4.4° E, 100 m | 1992–2016 | Noon irradiance at 307.5 nm | 0 | [56] |
| Hoher Sonnblick, Austria | 47.1° N, 12.9° E, 3106 m | 1997–2011 | Irradiance at 305 nm (SZA = 45–65°) | +5 to +8 (larger for larger SZA) | [143] |
| Hoher Sonnblick, Austria | 47.1° N, 12.9° E, 3106 m | 1997–2011 | Irradiance at 315 nm (SZA = 45–65°) | +9 * to +14 * (larger for larger SZA) | [143] |
| Thessaloniki, Greece | 40.6° N, 23.0° E, 60 m | 1994–2014 | Irradiance at 307.5 nm (SZA = 64°) | +5 * | [144] |
| Thessaloniki, Greece | 40.6° N, 23.0° E, 60 m | 1994–2014 | Irradiance at 350 nm (SZA = 64°) | +3 * | [144] |
| Thessaloniki, Greece | 40.6° N, 23.0° E, 60 m | 1992–2016 | Noon irradiance at 307.5 nm | +3 * | [56] |

[1] Statistically significant trends at the 95% confidence level are denoted with an asterisk (*).

Analysis of spectral UV records from four different stations at latitudes between 55 and 70° N in North America and Europe (Barrow, Alaska; Churchill, Canada; and two stations in Finland: Sodankylä and Jokioinen) for the period 1994–2011 showed a (statistically significant at the 90% confidence level) decrease of −3.9% per decade for the average daily irradiance at 305 nm while the irradiance at 325 nm remained stable during the same period [55]. The former was attributed to a corresponding significant trend of +1.4% per decade of the zonally averaged total ozone. Negative trends of the UV-B irradiance in April and June (10% per decade for 305 nm), which however are not statistically significant, have also been reported for Sodankylä, Finland by Lakkala et al. [140] for 1990–2014. The UV-A irradiance did not change during the same period, indicating that the trends in UV-B irradiance were driven by changes in total ozone. In a more recent study [56], analysis of spectral measurements for Sodankylä for the period 1992–2016 revealed a decreasing trend of −3.2% per decade in the irradiance at 307.5 nm, which again is significant at the 90% (and not the 95%) confidence level. Part of this decrease was attributed to a statistically significant decrease of 4.6% per decade in surface reflectivity. A non-significant increase of 0.8% per decade in total ozone was found for the same period. Analysis of the total ozone and UV irradiance measurements in different Norwegian stations from the Norwegian Institute for Air Research (NILU) for 1995–2016 shows that the average daily erythemal doses have decreased with an average rate of 2–5% [141], with the changes being again not significant. Not significant—positive in this case—trends of the annual average daily erythemal dose for the period 1996–2016 have also been reported for the arctic Polish station of Hornsund, Svalbard. The aforementioned studies show that changes in aerosols do not affect importantly the long-term changes of the UV radiation at the particular polar and sub-polar sites. The dominant role of ozone and clouds has been also discussed in another recent study regarding the variability of UV radiation in the high latitude site of Tõravere, Estonia (58.3° N, 26.5° E, 70 m a.s.l.) for 2004–2016 [145]. In the latter study, the authors showed that changes in aerosols did not affect the mid-term changes of the UV irradiance significantly.

Determining the main drivers of changes at latitudes below 55° N is more complex, mainly because changes in the aerosol composition and load are more significant, especially at urban sites [35,58,146–148]. Krzyścin, et al. [149] used a very long record of broad-band measurements and studied the long-term changes of the erythemal irradiance at Belsk, Poland for the period 1976–2008 reporting positive trends of 6% in its annual average levels, which were mainly attributed to changes in aerosols. In a more recent study Czerwińska and Krzyścin [150] showed that erythemal irradiance over the latter site remains however relatively stable after the mid-2000s. In a study for Uccle, Belgium, decreasing attenuation by aerosols and clouds was found to be the main contributor to significant increasing trends of the average erythemal UV dose by 7% per decade in 1991–2013 [58]. Analysis of the spectral UV record for Uccle for the period 1992–2016 showed that the overall result of the statistically significant increase of the total ozone (2.6% per decade), the statistically significant decrease of total cloud cover (−2.2% per decade), and the statistically significant decrease of absorption by aerosols (−0.4 units of the aerosol index [151,152] per decade) is that the noon irradiance at 307.5 nm remained stable in the period of study [56].

In Thessaloniki, Greece, a significant positive change of 3.3% per decade was detected in the clear-sky irradiance at 350 nm, for the period 1994–2014, which was again attributed to the significant decrease of the AOD in the same period (−0.1 per decade for AOD at 320 nm). Despite the small (non-significant) increase of the annually averaged total ozone (0.8% per decade) for the same period at Thessaloniki, the increase of the 307.5 nm irradiance was even stronger (+4.5% per decade), possibly because aerosols at Thessaloniki are more effective on absorbing UV radiation at lower wavelengths [147]. Although AOD decreased with a relatively constant rate in the period 1994–2014, UV irradiance increased very fast from 1994 until the mid-2000, and then remained relatively stable. Similar behavior of the UV irradiance (i.e., UV irradiance increasing until mid-2000 and then decreasing) has been also reported for Uccle, Belgium [58], and Chilton, UK [62], as well as for a number of other sites around the world [57]. In the case of Thessaloniki, this behavior is mainly due to the changing

absorption efficiency of aerosols during the period of study [147], while in Chilton and Uccle it is a result of the combined effects of changes of different factors (clouds, total ozone, and aerosols).

Study of the available erythemal UV dose for Chilton, UK, showed a statistically significant increase of 10% per decade for the period 1991–2004, and then a statistically significant decrease of −13.5% per decade for the period 2004–2015 [62]. This behavior was mainly due to decreasing cloudiness in the former period and increasing total ozone in the latter. Analysis showed that changes in aerosol load did not practically affect the long-term UV changes at the particular site. The average change of the effective radiant UV exposure for the period 1995–2015 was −8% per decade [142]. Analysis of the UV record at Reading (located at a distance of a few kilometers from Chilton, UK) for 1993–2008 yielded a significant positive trend in the daily maximum erythemal irradiance (6.6% per decade), although no significant reduction in the total ozone data series was detected, suggesting a reduction in cloud cover during the midday period [60].

As in Chilton, the role of aerosols is less significant than that of total ozone, clouds, and surface albedo over several mid-latitude sites (e.g., high altitude sites in the Alps) [40,153,154]. Schwarz, et al. [155] showed that over the Austrian Alps, 90% of the cases for which erythemal irradiance exceeds its climatological levels correspond to below-climatological levels total ozone. Analysis of the spectral UV record of Hoher Sonnblick, Austria, for the period 1997–2011 revealed significant positive trends, of 5–7% per decade (depending on the SZA) in the average levels of the 305 nm irradiance, despite the significant increase of 1.9% per decade in total ozone during the same period. Stronger positive trends were found for wavelengths larger than 315 nm (9–14% per decade depending on the SZA) indicating that the positive long-term changes in the UV are mainly driven by factors other than the changes in total ozone.

*3.2. Update for Four Historical European Stations*

Updated analysis of the same dataset used in Fountoulakis et al. [56] for four of the European stations discussed in Table 1 (Reading, Uccle, Sodankylä, and Thessaloniki) for the years 1996–2017, shows that over all four stations the observed long-term changes cannot be justified solely by changes in total ozone. For three out of the four stations (Reading, Uccle, and Thessaloniki), the trends in UV are statistically significant, and are mainly driven by changes in aerosols, surface albedo, and clouds. The results of the analysis are presented in Figure 1. In the panels on the first row, the annual mean anomalies of the all-sky noon irradiance at 307.5 nm, and the corresponding linear trends, are presented. In the panels on the second row, the corresponding results for the ratio between the irradiance at 307.5 nm and 324 nm (from now on referred to as ratio) are presented. The annual mean anomalies and the trends for ozone are presented in the panels of the third row. At the bottom of each panel the average change per decade (in %) and the corresponding uncertainty are shown (in red). Statistically significant trends are marked with an asterisk.

Uccle is the only site for which a statistically significant increase of total ozone has been detected. Based on this result, a negative trend in the 307.5 nm irradiance and/or in the ratio would be expected [26,156,157]. However, the 307.5 nm irradiance was increasing significantly, by 5% per decade, while the ratio was not changing in the long-term. The 307.5 nm irradiance at Thessaloniki also increased significantly by 8% per decade, despite the absence of any trend in total ozone. The 307.5/322.5 nm ratio also increased (though not significantly) for Thessaloniki. As has been thoroughly discussed in previous studies [56,58,144], changes in aerosols were responsible for a large part of the long-term changes in the UV irradiance over both sites. In urban environments, absorption in the UV-B region can be much stronger than the absorption in the UV-A [147,158], which could lead to more significant increase of the irradiance at UV-B wavelengths when aerosol load decreases. This behavior is associated with the presence of brown carbon in the urban aerosol mixture [159–161]. Assuming that this hypothesis is true, the stronger effect of the aerosol decrease on the UV-B relative to UV-A irradiance in Uccle is suppressed by the opposing effect of the total ozone increase, leading to similar

long-term changes for both the 307.5 and the 324 nm irradiance. However, further investigation, which is out of the scope of the present study, is necessary in order to confirm this assumption.

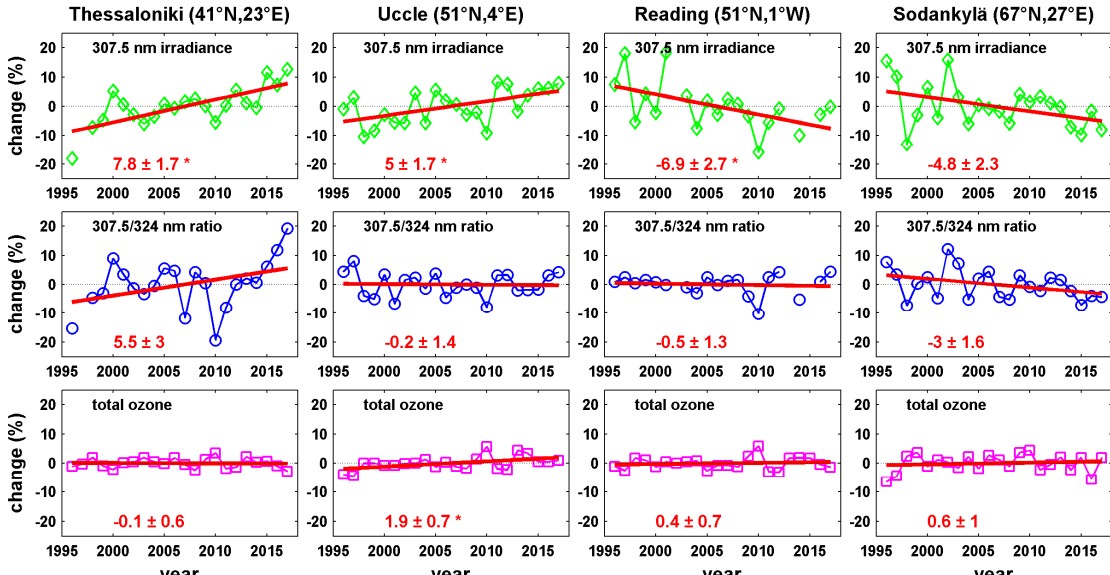

**Figure 1.** Annual mean anomalies of the all-sky noon irradiance at 307.5 nm (**upper panels**), the ratio between the all-sky noon irradiance at 307.5 nm and 324 nm (**middle panels**), and the total ozone (**lower panels**) over Thessaloniki, Greece (**first column**), Uccle, Belgium (**second column**), Reading, UK (**third column**), and Sodankylä, Finland (**fourth column**). Red lines represent the linear trends for the full period. The average (%) change per decade and the corresponding uncertainty are provided in red at the bottom of each panel. Asterisk denotes statistically significant changes. Updated from Fountoulakis et al. [56].

Negative trends of the 307.5 nm irradiance of about −7% and −5%, have been found for Reading and Sodankylä respectively (significant only for Reading), despite the absence of significant trends in total ozone. In the case of Reading changes in cloudiness possibly play a significant role [62], while for Sodankylä changes in surface albedo due to negative trends in snow cover in late winter and spring [162] may also be important as has been also discussed in previous studies [56]. It is quite interesting that strong, significant trends toward different directions have been detected for Uccle and Reading, despite the relatively short distance (less than 400 km) between the two sites. Given that the good quality of the spectral measurements at both sites (as well as the sites of Thessaloniki and Sodankylä) has been ensured through strict calibration and QA/QC procedures, this latter result is indicative for the large spatial variability of the mechanisms controlling the long-term changes of the solar UV irradiance.

## 4. Spectral UV Measurements in Italy

### 4.1. State of Measurements in Italy

Italy lies at the shores of the Mediterranean Sea, and extends in a wide latitudinal zone from 35 to 47° N. The topography of both the islands and the mainland is complex. The Alps at the North, and the Apennines extending from the north to the south, compose a very complex terrain with large altitude gradients even within horizontal distances of a few kilometers. Large altitude gradients exist even in the big islands at the southern part of the country. Thus, the meteorological conditions vary significantly throughout Italy, resulting in a correspondingly large spatial variability in the levels and the spectral distribution of the solar UV irradiance that reaches the surface [163]. During spring and summer the UV index may reach extremely high values in Italy, especially at the Alpine regions of the North, where extreme values up to 12 have been recorded [164]. At higher altitudes, the complex,

highly reflective snow-covered terrain significantly enhances the upwelling UV radiation, which in turn induces exposure rates (i.e., the ratio between the erythemal irradiance received by a person and that measured on a flat terrain in a particular place) above 1 [165]. In addition to skiers and sunbathers, people occupied in outdoor activities all over Italy are also highly exposed to UV radiation [166–168], and are subjected to increased risk for both melanoma [169] and non-melanoma [170] skin cancers. Exposure of the people living in Italy to solar UV radiation is however also beneficial, and associated with reduced incidence of diseases related to inadequate exposure to solar UV radiation compared to the inhabitants of Northern-European countries [171]. From the above discussion it follows that in Italy, accurate and timely information of the local population as well as tourists regarding the levels of solar UV radiation is absolutely necessary in order to achieve optimal sun-exposure behaviors, which will permit reaping the highest possible benefit from the exposure to solar UV radiation.

Manara et al. [172] showed that between 1986 and 2013 the total solar radiation increased in Italy, mainly as a result of changes in aerosols. The increase at the North (+7.7% per decade) is stronger than the increase at the South (+6% per decade). In a more recent study [173] focusing on the region of Piedmont (North Italy), an average increase of 2.5% per decade in the levels of the total solar irradiance, is reported for 1990–2016, resulting from changes in aerosols and clouds. In the latter study, it is also discussed that the trends vary temporally and spatially, depending on the season and the altitude of different sub-regions of Piedmont. In the future, UV irradiance is projected to increase over Italy, mainly as a result of improved air quality and reduced attenuation by clouds over the area [41,66,174,175]. Despite the significance and the possible impacts of changes in solar UV irradiance, and the availability of UV irradiance measurements since the beginning of the 1990s, there is currently no study focused on its long-term changes over the last decades.

Monitoring of the spectral global UV irradiance in Italy began in 1992, when two Brewer (MKIV type) spectroradiometers were installed at Rome and Ispra. More specifically, the Brewer#066 was installed at the Environment Institute of the European Union-Joint Research Centre, Ispra (45.8° N, 8.6° E, 240 m a.s.l.), while the Brewer#067 was installed at the Physics Department of Rome University "La Sapienza" (41.9° N, 12.5° E, 60 m a.s.l.). Since 1997, spectral UV measurements are also performed at the Station for Climate Observations of the Ente per le Nuove Tecnologie, l'Energia e l'Ambiente (ENEA), located at the island of Lampedusa (35.5° N, 12.6° E, 50 m a.s.l.). In 2007 the Brewer#067 was moved from Ispra to the headquarter of ARPA Valle d'Aosta (Aosta Valley Regional Environmental Protection Agency) at Saint Christophe, Aosta (45.7° N, 7.4° E, 570 m a.s.l.), where spectral UV measurements are also performed by a Bentham (type DTMc300) spectrometer (commercially available by Bentham Instruments Ltd, with headquarters at Reading, United Kingdom) since 2004. All instruments described above are calibrated against reference standards on a regular basis, providing in this way, reliable, high quality spectral UV measurements throughout the whole period of their operation. Continuous narrow-band and/or broad-band UV measurements are currently also available from filter radiometers at Lampedusa and Rome, as well as from a few more Italian stations located at different sites such as Bologna, Vicenza, Florence, and elsewhere [107,176].

## 4.2. Climatological Analysis for Rome and Aosta

As discussed in Meloni et al. [163], the available erythemal doses throughout Italy vary temporally and spatially as a function of latitude, altitude, and time of the year. In the following, the monthly climatologies of the noon UV index at Rome and Aosta, two sites at quite different latitudes (the latitudinal difference between Rome and Aosta is 4°) and environmental context were compared to each other (Figure 2), in order to show how the environmental conditions at each of the two sites affect the spectral UV irradiance, and in turn the noon UV index over different months of the year.

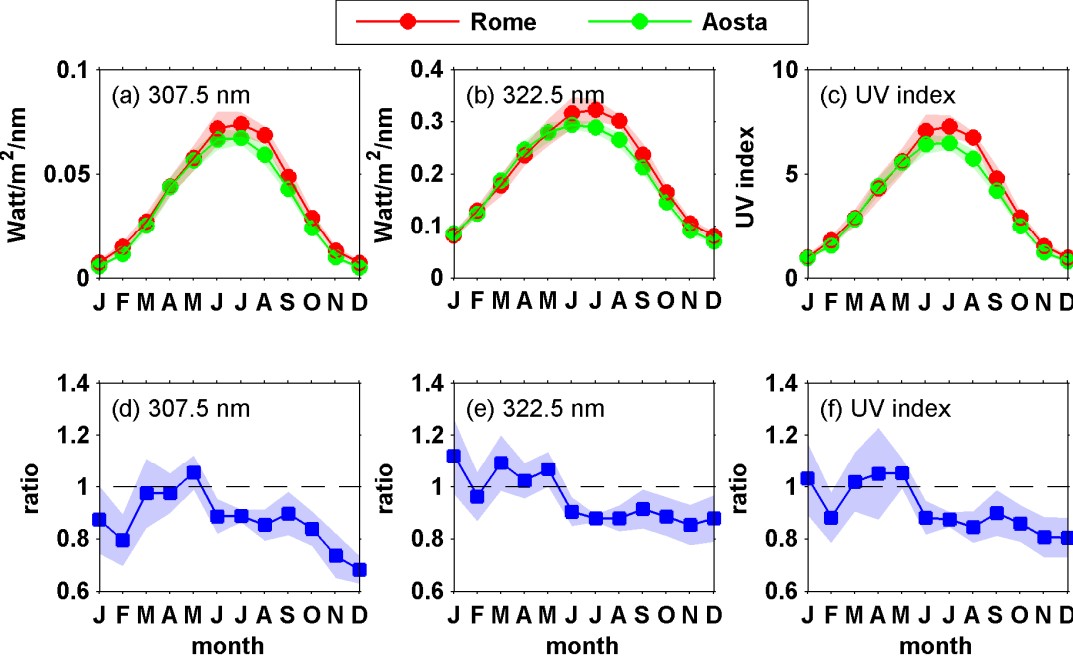

**Figure 2.** Climatologies (**a**–**c**) of the irradiance at 307.5 nm, 322.5 nm, and the UV index and the corresponding average ratios (**d**–**f**) between the two stations (Aosta/Rome) for the period 2006–2015. Shaded envelopes correspond to the standard deviation of the climatological values and the average ratios.

The climatological values of the irradiance at 307.5 and 322.5 nm, and the noon UV index are presented in panels a–c, and the corresponding average ratios between measurements in Aosta and Rome for each quantity are shown in panels d–f. The total ozone climatology is presented in Figure 3. Shaded areas represent the standard deviation of the averages. Climatological values of the TCAF, the surface albedo, and the AOD at 550 nm are presented in Figure 4.

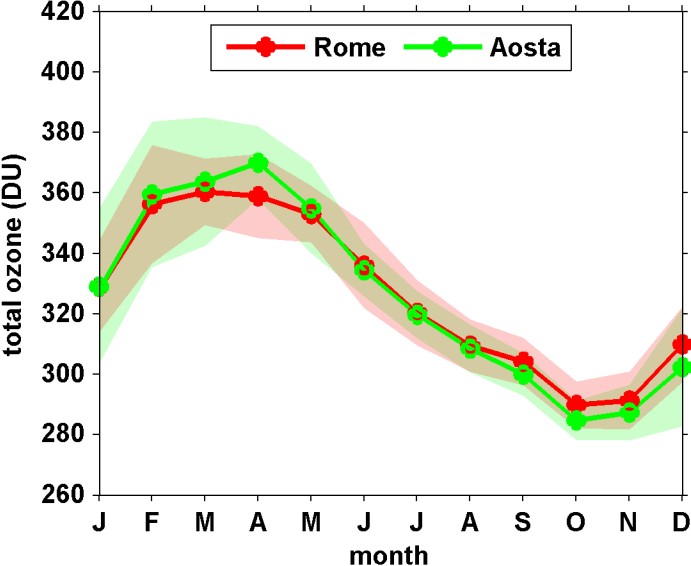

**Figure 3.** Monthly climatology of the daily average total ozone for Rome and Aosta for the period 2006–2015. Shaded envelopes correspond to the standard deviation.

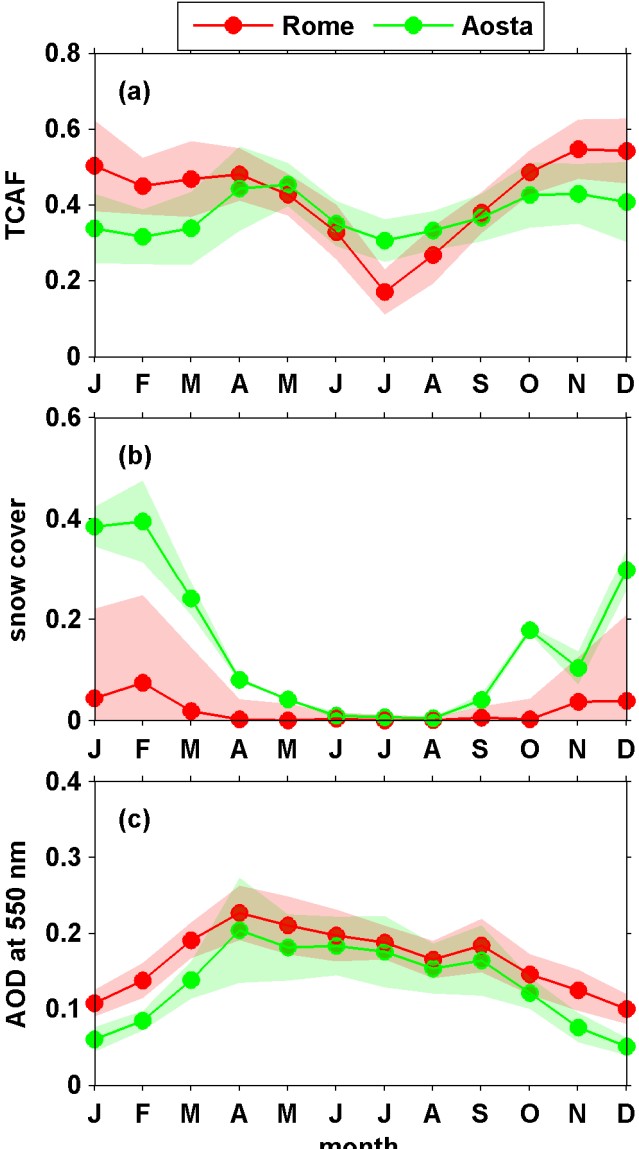

**Figure 4.** Monthly climatology of (**a**) the total cloud area fraction (TCAF), (**b**) the snow cover, and (**c**) the aerosol optical depth (AOD) at 550 nm, for Rome (**red**) and Aosta (green) for the period 2006–2015. Shaded envelopes correspond to the standard deviation limits.

The annual variability of the total ozone was slightly larger in Aosta than in Rome, which is reasonable given that Aosta is at higher latitude [177]. During the period September–December total ozone over Aosta was 5 DU below the total ozone over Rome, while in spring the total ozone was higher over Aosta, with a maximum difference of 10 DU in April. As discussed in the following, these differences alone do not justify the differences in the levels of the UV irradiance at the two sites.

At Rome, the UV index, as well as the irradiance at 307.5 nm and 322.5 nm became maximum in July instead of June (when the noon SZAs were at their minimum), mainly because of less attenuation by clouds in July relative to June. However, in Aosta the UV index and the 307.5 irradiance in June and July were nearly equal, while the 322.5 nm irradiance was slightly higher in June than it is in July. Since the total ozone at both sites was nearly the same in June and July, this was possibly because in July clouds attenuate UV irradiance more effectively in Aosta than in Rome, as shown in Figure 4.

In January and February, the 322.5 nm irradiance at Rome differed by less than 10% from the corresponding quantity for Aosta. The higher altitude of the Aosta site [39] in conjunction with the presence of less clouds, less aerosols, and the more reflective terrain relative to Rome, seemed to

compensate for the effect of smaller noon SZA for the particular wavelength. The 307.5 nm irradiance in Aosta for the same months was 20% lower than in Rome, although the total ozone was practically the same. The main reason for the latter result was that the latitudinal difference of 4° between the two sites induces a corresponding difference in noon SZA. This difference became more significant in winter and led to larger differences in the optical path of the irradiance. Increasing the optical path has a stronger impact on lower wavelengths, which are scattered from the atmospheric molecules and aerosols, and absorbed by the total ozone, more effectively than the irradiance at longer wavelengths. It should also be taken into account that the Aosta monitoring station was at the bottom of the Aosta Valley, surrounded by high mountains, which cover part of the horizon, and block part of the scattered light reaching the sensor of the instrument. The fraction of the scattered light, which was blocked by the mountains, was more significant for the irradiance at shorter wavelengths. As a result of all the above, the UV index was nearly equal at the two sites in January, and 10% lower at Aosta in February.

In December, the noon SZAs over each of the two stations were minimal relative to the rest of the year. Thus, the difference between the optical paths of the irradiance reaching the two sites became more significant. The fraction of the scattered UV radiation that was blocked by the very tall mountains south of the Aosta monitoring site also became maximum relative to the rest of the year. This is probably why the UV index in Aosta was 20% lower than the UV index in Rome, although attenuation of the solar UV radiation from clouds, total ozone, and aerosols was weaker at Aosta. Although snow cover (and consequently surface albedo) was high and enhanced the levels of the UV irradiance at the surface, it was lower than in January. All the above explain the very low ratios of 0.85 and 0.65 for the irradiances at 322.5 nm and 307.5 nm respectively in the particular month.

In autumn and spring the differences in the total ozone, surface albedo, AOD, and TCAF became less significant relative to winter, especially in May and September. Even though the magnitude of the differences between the two sites regarding the main factors affecting UV radiation in spring was very similar to autumn, the ratio between the irradiances was 1 or higher in spring, and 0.9 or lower in autumn. The slightly higher SZAs in spring were not enough to explain the different ratios, which might be due to differences in the types of clouds and aerosols over the two sites, and/or the slightly higher albedo in spring relative to autumn in Aosta.

Noon UV irradiance at both wavelengths, and the UV index, should be nearly the same at the two sites during summer if there were no differences in surface albedo, AOD, and TCAF, since the difference in the noon SZA became less important. Clouds however attenuated a larger fraction of the solar radiation over Aosta relative to Rome, especially in July, being (at least partially) responsible for the fact that UV irradiance (erythemal, and at 307.5 and 322.5 nm) at Aosta was 15% less than at Rome.

The effect of differences in cloudiness, AOD, and surface reflectivity became clearer if the irradiances at the two sites were compared for a particular SZA instead of the local noon. Thus, the comparison was performed for SZA = 65° (average of the ratios for SZAs 63–67° as described in Section 2) and its results are presented in Figure 5. The modeled ratios, when only differences in altitude and total ozone were taken into account, were also presented in the same figure.

The nearly constant modeled ratio of 1.05 for the irradiance at 322.5 nm (Figure 5b) was a result of the higher altitude of Aosta. The effect of altitude was slightly larger on the 307.5 nm irradiance. For January–February and June–August, climatological total ozone in Rome and Aosta differed by less than 2 DU, and the modeled ratio was 1.07–1.08 (Figure 5a). The modeled ratio for 307.5 nm decreased to 1 in April and increased to 1.15 in autumn and early winter, as a result of corresponding differences (5–10 DU) in the total ozone.

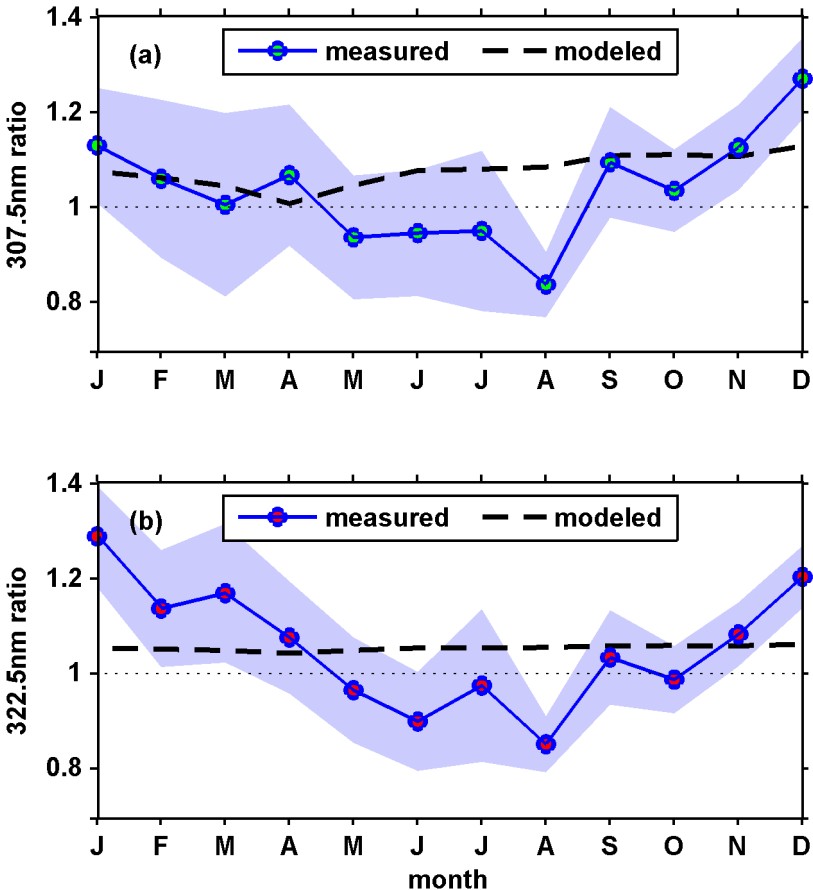

**Figure 5.** Average ratios between the monthly average irradiance at (**a**) 307.5 nm and (**b**) 322.5 nm measured in Aosta and Rome at SZA = 65° for the period 2006–2015. Shaded envelopes correspond to the standard deviation of the climatological values and the average ratios. The blue doted lines represent the theoretical ratios when only differences in altitude and climatological ozone are taken into account.

There was an evident annual cycle (in Figure 5) in the ratios for both, 307.5 and 322.5 nm, which practically followed the annual course of the differences in cloudiness and surface albedo (Figure 5). The differences between the modeled and the measured ratios were nearly the same for the two wavelengths for months April to December. The 322.5 nm ratio was slightly higher in the first half of the year relative to the second half, exceeding significantly the modeled values in months December–March. During the period January–March, the 307.5 nm ratio was lower by 0.15 relative to the 322.5 nm ratio and was very close to the modeled values. A more comprehensive investigation of the spectral characteristics of the attenuation by clouds and aerosols at the two sites was necessary in order to explain this result. Part of this difference could be explained by the fact that the reflected UV-B irradiance was absorbed more effectively than UV-A by ozone in the atmosphere. In summer, the effect of stronger attenuation by clouds over Aosta was dominant over the effect of stronger attenuation by aerosols in Rome, leading to lower measured ratios by 0.1–0.25 relative to the modeled values.

### 4.3. UV from Forecast Models and Satellite Retrievals

From the discussion of Section 4.1 it is clear that the levels of the solar UV irradiance at the two sites were controlled by very complex processes involving a number of different parameters. Thus, it is difficult to accurately incorporate these parameters in radiative transfer codes (such as those used for UV forecast models or satellite UV estimates) and model spectral UV irradiance, and consequently

human-health related effective doses. In this section, the UV index measured at the local noon in Aosta and Rome was compared with estimates from the DWD model and the OMI algorithm.

In Figure 6, the differences between the UV index from the DWD model and ground-based measurements (DWD-GB), and between OMI retrievals and ground-based measurements (OMI-GB), are presented in red and blue respectively. The results are presented separately for Aosta (a) and Rome (b). The corresponding smoothed averages of the differences for a ±15 days window are presented in panels (c) and (d).

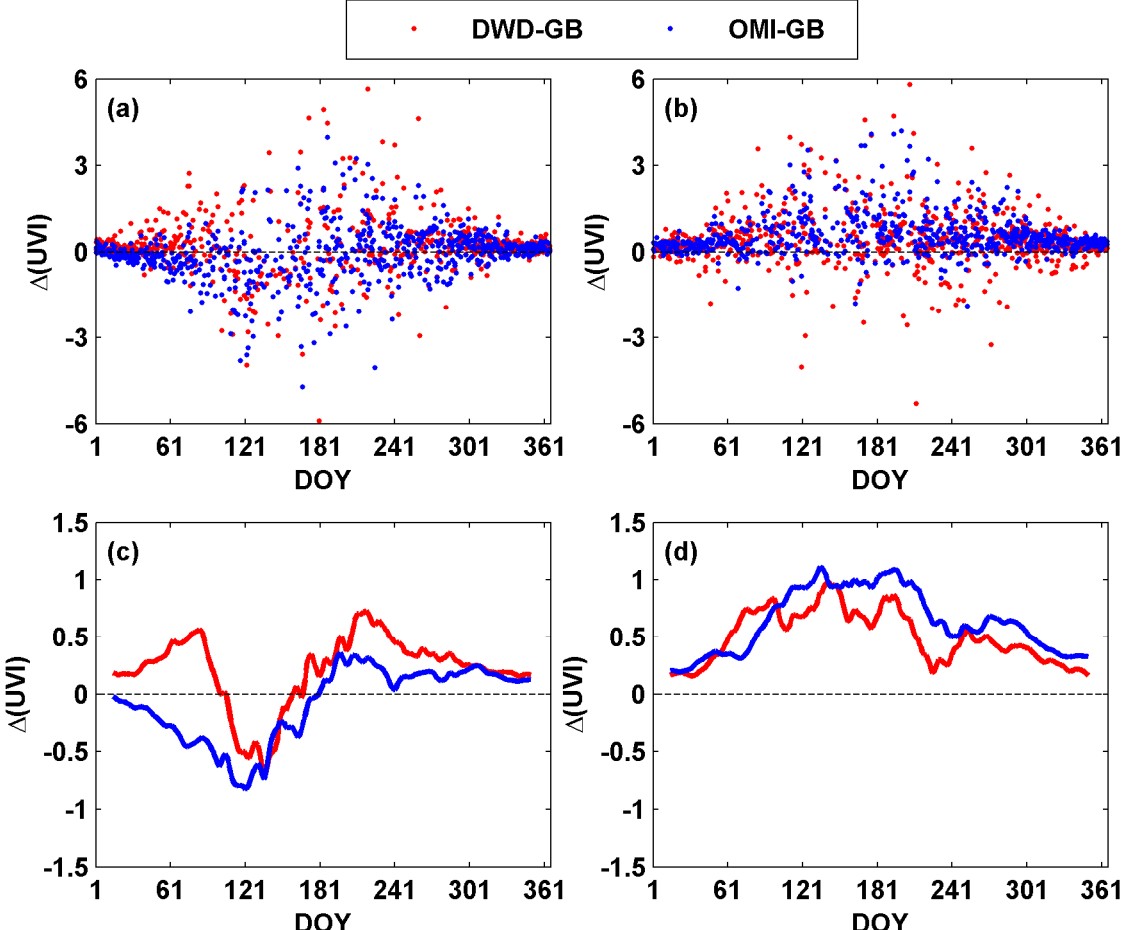

**Figure 6.** Difference between the noon UV index from ground based measurements and Deutscher Wetterdienst (DWD) model (red dots), and the noon UV index from ground based measurements and ozone monitoring instrument (OMI; blue dots) for (**a**) Aosta and (**b**) Rome. The smoothed averages for a ±15 days moving window are presented in the lower panels for (**c**) Aosta and (**d**) Rome. Red and blue lines correspond to red and blue dots respectively. For clarity, the scale of the y-axis is different at the upper and lower graphs.

About 5% of the DWD estimates for Rome and 8% of the DWD estimates for Aosta differ from the measured UV index by more than 1 unit. In extreme cases, differences up to 6 units of the UV index were found. Very large differences were most likely due to the different effect of clouds on the average irradiance over the pixel of the model, and the irradiance at the two stations, as well as uncertainties in the modeling of clouds [178]. The corresponding percentages for the differences in ground-based measurements of UV index to OMI estimates were 2% and 10%, respectively, which again could be justified by the fact that ground-based measurements refer to a particular point, while the dimension of the OMI picture at nadir was $13 \times 24$ km$^2$, reaching $13 \times 100$ km$^2$ at the edges of the swath. Thus, in particular cases clouds had a completely different effect on the two quantities

(ground-based observations versus OMI estimates) leading to larger differences. This effect is more important over the Alps, where clouds may induce large variability of the irradiance within the satellite pixel [87]. Analyzing only the data corresponding to cloud fraction below 40% in the satellite pixel reduced significantly the number of points for which the difference between ground-based and satellite (or modeled) data exceeded 50% of the measured UV index, verifying that very large differences were mainly due to the different effect of clouds in the data from different sources.

On average, the UV index from both the DWD model and OMI observations was overestimated over Rome, with the overestimation generally being stronger in the period April–July. This result was in agreement with the findings of Marchetti, et al. [179] who compared the UV index from OMI with measurements from the station of Rome. As for most mid-latitude, low-altitude stations [180], the total ozone measured by OMI at Rome differs from ground based measurements by less than 1% for the particular period, which is in agreement with the results of Ialongo et al. [181], and cannot explain the differences in the UV index. The most probable explanation is that the absorption of the UV radiation by aerosols at Rome is underestimated in the DWD and OMI algorithms [182], leading to the calculation of higher UV index [67] through the whole year. In summer the UV index became larger due to smaller noon SZAs, and the absolute difference between the OMI and the ground-based measurements increased. However, investigation of the % relative—and not the absolute—difference between the UV index measured at the surface and the UV index from OMI and DWD revealed larger relative differences in winter. Although aerosol load is generally larger in spring and summer at Rome [127], this result may indicate that the different composition of aerosols in different seasons of the year [183,184] may be responsible for more effective absorption of the UV irradiance in winter than in summer.

The patterns of the average differences between the measured and modeled (by DWD or OMI data) UV index for Aosta were different from those at Rome (Figure 6). The DWD model generally overestimated the UV index, with the exception of April and May. OMI underestimated the measured UV index during the first half of the year, with a peak underestimation in April and May, and overestimated during the second half of the year. The role of aerosol in Aosta was not as significant as in Rome, although under particular conditions highly absorbing aerosols may either be transported to the Aosta Valley or produced locally [185,186]. The absorbing effect of these aerosols cannot be easily described by the model or the satellite algorithm, and possibly is responsible for, at least part of, the overestimation of the UV index in summer, autumn, and winter. Nevertheless, the spatial variability of aerosols and other pollutants within the pixel of OMI also affects the results of the comparison for both, Rome and Aosta [92]. The patterns and the magnitude of the average differences presented in panels (c) and (d) of Figure 6 did not change significantly when only data corresponding to cloud fraction below 40% were used for the analysis.

Comparison between the total ozone from OMI and ground-based measurements for the same period shows that the total ozone over Aosta was underestimated on average by 7 DU (2%). This underestimation may be due to a corresponding underestimation of tropospheric ozone by OMI [187], with the underestimation being maximum in April and May. In other Alpine sites, maximum tropospheric ozone levels have been reported for the particular months [188], enhancing the above assumption. In contrast to what would be expected (due to the underestimation of total ozone), OMI, as well as the DWD model, underestimate the UV index in spring. At this point it has to be referred that at Aosta the agreement of total ozone from ground-based measurements was better with the total ozone from the DWD model than the total ozone from OMI (average difference was −1.4 DU for the former and −7 DU for the later). This was not the case for Rome, where the corresponding differences were +5 DU and −1 DU.

Given that differences in total ozone could not explain the differences in the UV index from the three different sources, the under- or overestimation of the UV index from the DWD model and OMI could be attributed to the inaccurate description of the effects of surface albedo, aerosols, and cloudiness. In the study of Wagner et al. [87] underestimation of the UV index by OMI over several

mountainous sites is attributed to erroneous cloud correction rather than the inaccurate description of surface albedo. Nevertheless, UV index from OMI and the DWD model refers to the average altitude of the pixel, which certainly differs from the exact altitude of the station.

## 5. Summary and Conclusions

Since the 1990s, the levels of the spectral UV irradiance have changed over Europe, mainly as a result of changes in aerosols, clouds, and surface albedo. All studies referring to stations in South, East, and Central Europe report positive trends of the UV irradiance, mainly resulting from reduced attenuation by aerosols and clouds. A number of studies for different stations in the UK report decreasing UV irradiance in the last three decades. Significant trends of −7 to −8% per decade (in the erythemal UV dose and the irradiance at 307.5 nm) have been detected for the period between the mid-1990s until the mid-2010s in Reading and Chilton, UK. However, further analysis for the latter station showed that UV irradiance was increasing up to 2004 and decreasing thereafter. For the same period, significant positive trends of the same magnitude have been detected for Uccle, Belgium, which is within a distance less than 400 km from Reading and Chilton. Analysis of the UV datasets for the two stations in the context of the present study confirmed these findings and showed that UV-B and UV-A irradiances at the particular stations change with the same rate (possibly for different reasons) during the period 1996–2017. The big difference between the results for Uccle and Reading is a typical example of the large variability of changes in aerosols and cloudiness, which subsequently result in large spatial variability in the trends of UV irradiance. A large significant increase of 8% has been found for the irradiance at 307.5 nm at the station of Thessaloniki for 1996–2017, despite the absence of significant trends in total ozone or cloudiness at the particular station [56,189]. This trend has been mainly attributed to changes in aerosols.

Studies referring to higher latitude stations report decreasing UV irradiance. A positive, not significant trend of the average daily erythemal dose has been reported for the arctic station of Hornsund, Svalbard for the period 1996–2016. Analysis of the UV time-series for different stations in Norway, as well as for Sodankylä, Finland, reveals negative trends in the erythemal irradiance and the irradiance at different UV-B and UV-A wavelengths, ranging between 2% and 5%. However, these trends are not statistically significant. Analysis in the context of the present study shows that in the period 1996–2017 the UV irradiance at 307.5 and 324 nm decreases in Sodankylä, with the trends being non-significant and of similar magnitude as those reported from previous studies. Lakkala et al. [140] reported larger but insignificant trends, of the order of 10% for the period 1990–2014 in the monthly average levels of the irradiance in the UV-B region in spring and the early summer, mainly resulting from increasing ozone. At higher latitudes, part of the changes in UV can be attributed to changes in surface albedo and cloudiness.

In Italy several studies report significant changes in cloudiness during the last decades [172,173], which are expected to have an impact on the levels of UV irradiance. Although, continuous spectral UV measurements of good quality are available from different stations, the long-term changes of UV irradiance have not been investigated yet. The inhomogeneous terrain throughout the Italian territory and the large altitudinal gradients in particular regions, lead to large differences between the factors controlling the variability of UV irradiance even within nearby sites. Differences between the daily minimum SZA were also significant since Italy is extended in a long latitudinal zone. In order to discuss these effects, the annual variability of the irradiance at 307.5 and 322.5 nm and the erythemal irradiance were investigated for the urban, low altitude site of Rome in central Italy, and the alpine site of Aosta at the northern borders of the country.

Analysis of satellite data shows that in Rome the attenuation of UV radiation by aerosols was more important than in Aosta, throughout the whole year. Attenuation by clouds in months October–March was stronger at Rome, while in the summer (June–August) attenuation by clouds was stronger at Aosta. UV irradiance at Aosta in months October–April was enhanced by the high surface albedo. The total ozone was higher by 5–10 DU in Aosta in spring and by 5–10 DU in Rome in autumn. When

the differences between the irradiance over the two sites were studied for a particular SZA (=65°) the differences in surface albedo and cloudiness were clearly depicted in the results, while the effect of differences in total ozone was less significant. The ratio between the irradiances at Aosta and Rome (for both, 307.5 and 322.5 nm) ranged from 0.8 in summer to 1.25 in winter. Theoretical analysis shows that at SZA = 65° the ratio would be 1.05 for 322.5 nm and 1.08 for 307.5 nm due to the higher altitude of Aosta, assuming that all other factors were constant.

When the irradiances measured at local noon at the two sites were compared to each other, the effect of different SZA became important and more pronounced for the 307.5 nm. The ratio for 307.5 nm ranged from a minimum of 0.7 in December to a maximum of 1.05 in May. Minimum and maximum values of the ratio were found for the same months for 322.5 nm and the UV index. Nevertheless, the variability is smaller, (i.e., 0.85–1.1 for the 322.5 nm irradiance and 0.8–1.05 for the UV index). The climatological noon UV index in Rome in July was 7.3 and was larger than the UV index in June (7.0), although the noon SZAs were smaller in June. The reason was possibly that clouds attenuate more UV radiation in June than in July. At Aosta, increased cloud cover in June relative to July counterbalanced the effect of SZA less effectively than in Rome, leading to nearly the same value of the UV index equal to 6.6 for both months.

Comparison of the noon UV index measured at the sites of Rome and Aosta for a four year time-period with the corresponding UV index from OMI and DWD forecast model estimates, further highlighted the significance of having accurate ground-based UV measurements. Large differences up to 6 units were found between ground-based measurements and both the DWD model and OMI for particular days. These differences resulted from an inaccurate description of the attenuation by clouds by both model and satellite estimates. Predictions of the current DWD model algorithm refer to a pixel of $13 \times 13$ km$^2$ instead of a particular point like in are the ground-based measurements. The OMI pixel refers to an even larger area of $13 \times 100$ km$^2$, wherein cloudiness is again not necessarily representative for Rome or Aosta. The use of climatological values for aerosol absorption properties, in both the OMI algorithm and the DWD model, which in many cases underestimated absorption by aerosols in the UV, possibly leads to the average overestimation of the UV index by 0.5 in winter to 1 in summer over Rome. In Aosta, the UV index was mainly overestimated in the second half of the year, possibly as a result of the inaccurate description of the aerosol absorption in the model and the satellite algorithm, and the underestimation of total ozone by OMI. Underestimation of the UV index in the first half of the year, mainly from OMI, could be due to the systematic overestimation of the attenuation by clouds and underestimation of the effect of surface albedo.

The comparison between the levels of the UV irradiance at the stations of Rome and Aosta revealed the significant role of aerosols, clouds, and surface albedo in the formulation of the levels of the spectral UV irradiance at each site. Changes in the particular parameters in the future might induce large changes in the levels of the UV irradiance that reaches the Earth's surface. Study of the trends of spectral UV irradiance in the past, with respect to the changes of the latter parameters and total ozone might provide useful information on how the spectrum of UV irradiance changed in the past decades. Investigation of the interactions between spectral UV irradiance and the factors discussed above might also provide very useful information, which would contribute to the improvement of the modeling of the UV irradiance.

The present study focused on pointing out the significance of having long-term, accurate UV measurements in Italy, in the context of changes in air quality and climate. Thus, scientific questions arising from the present study were in many cases not fully investigated and remained unanswered. Extension of the present study could include validation of different satellite and forecast UV products for a longer time-period with respect to total and tropospheric ozone, clouds, snow cover, and aerosol optical properties, since ground-based measurements of all the above parameters are available for Rome and Aosta. Development and application of new algorithms in the European SkyNet Radiometers Network (EUROSKYRAD) network is expected to give new products in the near future, which will contribute significantly in the understanding of the complex interactions between aerosols and solar

UV radiation. Total solar radiation measurements and sky camera products, which are available for both stations could also provide useful information for the role of clouds. Analysis of the results from more UV monitoring stations in Italy, in addition to those of Rome and Aosta, is necessary in order to investigate the spatial distribution of changes throughout the country.

In any case, the results of the present study clearly demonstrated the importance of maintaining and analyzing long-term, continuous, high quality ground-based measurements.

**Author Contributions:** Conceptualization, I.F., H.D., A.-M.S.; methodology, I.F., H.D., A.-M.S and C.S.Z.; software, I.F.; validation, I.F., H.D., A.-M.S., K.G., V.D.B., K.L., A.R.W., M.-E.K. and T.K.; formal analysis, I.F. and H.D.; investigation, I.F. and H.D.; resources, H.D., A.-M.S., G.L., A.F.B., H.D.B., K.L., T.K., A.R.W. and A.A.; data curation, I.F., H.D., A.-M.S., G.L., G.F., M.-E.K., K.G., V.D.B., K.L., T.K. and J.K.; writing—original draft preparation, I.F.; writing—review and editing, H.D., A.-M.S., G.L., G.F., A.A., M.-E.K., K.G., A.F.B., V.D.B., H.D.B., K.L., T.K., A.R.W., J.K. and C.S.Z.; visualization, I.F.; supervision, H.D.; project administration, H.D.; funding acquisition, H.D. All authors have read and agreed to the published version of the manuscript.

**Funding:** The spectral UV and ozone measurements at reading, UK were funded by UK Department of Environment, Food and Rural Affairs (DEFRA). The Academy of Finland has funded the UV measurements at Sodankylä by the FARPOCC and SAARA projects.

**Acknowledgments:** The ground-based data used in this publication were obtained as part of the WMO Global Atmosphere Watch publicly available via the World Ozone and UV Data Centre (http://woudc.org), and from the EUVDB (http://uv.fmi.fi/uvdb/) database. We would like to acknowledge and warmly thank all the investigators that provide data to these repositories on a timely basis, as well as the handlers of these databases for their upkeep and quality guaranteed efforts. The QBO indexes are from the CDAS Reanalysis data and are the zonally averaged winds at 30 and 50 hPa and taken from over the equator (http://www.cpc.ncep.noaa.gov/data/indices/). The F10.7 cm solar radio flux density, used as proxy of the 11-year solar cycle, was acquired from the Solar and Terrestrial Physics Division (STP) of NOAA's National Geophysical Data Center (NGDC) Website (http://www.ngdc.noaa.gov/stp/stp.html). The authors acknowledge the NASA GES-DISC Interactive Online Visualization and Analysis Infrastructure (GIOVANNI), developed and maintained by the NASA GES DISC, for providing MERRA-2 reanalysis, and OMI data. The Academy of Finland has funded the UV measurements at Sodankylä by the FARPOCC and SAARA projects.

**Conflicts of Interest:** The authors declare no conflict of interest.

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
