# Peer review of "Solar UV Irradiance in a Changing Climate: Trends in Europe and the Significance of Spectral Monitoring in Italy"

_environments, doi:10.3390/environments7010001_

Round 1

Reviewer 1 Report

Comments in the attached file

Reviewer 2 Report

This article is very well written and robust. The methods used are described in detail, and the methodology is sound, so I fully suggest publication of this study. I have 2 minor corrections, listed below, and a suggestion:

In the introduction (in the paragraph starting at line 124 especially), there’s a few very recent articles worth mentioning related to the future global and regional evolution of both ozone and UV radiation, using results from the Climate-Chemistry Model Intercomparison projects. I think they would be a valuable addition to the -already rich and well documented- discussion.

The corrections are:

Line 314 – is it 2006-2015 or 2006-2016? Both are used in the same paragraph Line 371 – have been downloaded (it’s referring to the daily averages)

References

Lamy, K., Portafaix, T., Josse, B., Brogniez, C., Godin-Beekmann, S., Bencherif, H., Revell, L., Akiyoshi, H., Bekki, S., Hegglin, M. I., Jöckel, P., Kirner, O., Liley, B., Marecal, V., Morgenstern, O., Stenke, A., Zeng, G., Abraham, N. L., Archibald, A. T., Butchart, N., Chipperfield, M. P., Di Genova, G., Deushi, M., Dhomse, S. S., Hu, R.-M., Kinnison, D., Kotkamp, M., McKenzie, R., Michou, M., O'Connor, F. M., Oman, L. D., Pitari, G., Plummer, D. A., Pyle, J. A., Rozanov, E., Saint-Martin, D., Sudo, K., Tanaka, T. Y., Visioni, D., and Yoshida, K.: Clear-sky ultraviolet radiation modelling using output from the Chemistry Climate Model Initiative, Atmos. Chem. Phys., 19, 10087–10110, https://doi.org/10.5194/acp-19-10087-2019, 2019.

Dhomse, S. S., Kinnison, D., Chipperfield, M. P., Salawitch, R. J., Cionni, I., Hegglin, M. I., Abraham, N. L., Akiyoshi, H., Archibald, A. T., Bednarz, E. M., Bekki, S., Braesicke, P., Butchart, N., Dameris, M., Deushi, M., Frith, S., Hardiman, S. C., Hassler, B., Horowitz, L. W., Hu, R.-M., Jöckel, P., Josse, B., Kirner, O., Kremser, S., Langematz, U., Lewis, J., Marchand, M., Lin, M., Mancini, E., Marécal, V., Michou, M., Morgenstern, O., O'Connor, F. M., Oman, L., Pitari, G., Plummer, D. A., Pyle, J. A., Revell, L. E., Rozanov, E., Schofield, R., Stenke, A., Stone, K., Sudo, K., Tilmes, S., Visioni, D., Yamashita, Y., and Zeng, G.: Estimates of ozone return dates from Chemistry-Climate Model Initiative simulations, Atmos. Chem. Phys., 18, 8409–8438, https://doi.org/10.5194/acp-18-8409-2018, 2018.

Author Response

We want to acknowledge Anonymous Reviewer#2 for his comments. We have replied to each of them in the following.

comment#1

In the introduction (in the paragraph starting at line 124 especially), there’s a few very recent articles worth mentioning related to the future global and regional evolution of both ozone and UV radiation, using results from the Climate-Chemistry Model Intercomparison projects. I think they would be a valuable addition to the -already rich and well documented- discussion.

reply

The suggested articles are now cited in the document (references#73 and #79)

comment#2

Line 314 – is it 2006-2015 or 2006-2016? Both are used in the same paragraph

reply

It was a typo. The period is 2006 - 2015. It has now been corrected.

comment#3

Line 371 – have been downloaded (it’s referring to the daily averages)

reply

corrected

Reviewer 3 Report

The paper Solar UV irradiance in a changing climate: Trends in Europe and the significance of spectral monitoring in Italy' by Fountoulakis et al. is well organised and has a clear construction. Methodology seems to be adequate to the considered problem and is clearly described. Results are provided in the form of understandable tables and graphs. The work aims to prove that ground-based monitoring is still very important, although there are new ways of estimating UV radiation, such as forecasts and satellite scanning. The role of aerosols is underestimated by models and satellites cannot take into account the variability of cloudiness in the local scale (especially over inhomogeneous terrain, such as mountains). Although the work is not original, it is still very much needed to make us aware of the importance of ground-based measurements.

General comments:

The paper is well written, with clear construction. The abstract summarize the work adequately, describes the overall work and contains results. The problem raised by the researches is of a great importance. The article presents the review of ground-based measurements and trend over Europe, which is very much needed and helpful for other researchers. The reviewer has only a few detailed comments, which may be helpful to the authors.

Detailed comments:

 Table 1, Page 8: There is a Polish station at Hornsund, but Hornsund is not a part of Poland, thus it table 1 should be Norway, not Poland;

Lines 504-506, Page 12: Have you considered the possible step change in Uccle data, that could generate the significant positive trend?

Author Response

We want to acknowledge reviewer#3 for his/her comments. We have replied to each of the comments below.

Comment#1

Table 1, Page 8: There is a Polish station at Hornsund, but Hornsund is not a part of Poland, thus in table 1 should be Norway, not Poland;

reply

corrected

Comment#2

Lines 504-506, Page 12: Have you considered the possible step change in Uccle data, that could generate the significant positive trend?

reply

Although we analyzed the data further, we were not able to identify a "jump" in the series which would justify a step change or any similar problem. Detection of such a problem was also not possible by comparing spectral measurements from different instruments operating at the station.

Nevertheless, the fact that Brewer#016 is calibrated on a regular basis, and every 2 - 4 years measures side by side with the reference Brewer#017 ensures the good quality of the data and reduces the possibility that such a problem occurred without having been detected.  

Some relative discussion has been also added in the manuscript.

Reviewer 4 Report

The paper raises the necessary and relevant topic. It is apparent that the authors went to considerable lengths to collect and analyze data. In my view, the big plus is that the technique is described in great detail, and the graphs are clear and understandable. However, the introduction is too long, there are many well-known facts, especially in the first part. In my opinion, the authors might write less about the effects of UV radiation on humans. The summaries are also quite voluminous. Trends are defined, but there are not enough clear explanations and their substantiation.

Author Response

We acknowledge Reviewer#4 for his/her comments. We have replied to each of them in the following.

Comment#1

The introduction is too long, there are many well-known facts, especially in the first part. In my opinion, the authors might write less about the effects of UV radiation on humans. 

reply

Most discussion regarding the effects of UV on humans has been removed from the introduction as the suggested by the reviewer.

Comment#2

The summaries are also quite voluminous. 

reply

Unfortunately it is not easy to drastically reduce the size of the text in paragraphs 3.1 and 4.1 (summaries for Europe and Italy respectively) without removing significant information. However, we removed part of the discussion in section 4.1. 

Comment#3

Trends are defined, but there are not enough clear explanations and their substantiation.

reply

We tried to improve the discussion in section 3.2 according to the reviewer's suggestion.